# Muscarinic Acetylcholine Receptor M3 Expression and Survival in Human Colorectal Carcinoma—An Unexpected Correlation to Guide Future Treatment?

**DOI:** 10.3390/ijms24098198

**Published:** 2023-05-03

**Authors:** Leonard A. Lobbes, Marcel A. Schütze, Raoul Droeser, Marco Arndt, Ioannis Pozios, Johannes C. Lauscher, Nina A. Hering, Benjamin Weixler

**Affiliations:** 1Department of General and Visceral Surgery, Charité—Universitätsmedizin Berlin, Corporate Member of Freie Universität Berlin and Humboldt-Universität zu Berlin, Hindenburgdamm 30, 12203 Berlin, Germany; 2Clarunis, Department of Visceral Surgery, University Centre for Gastrointestinal and Liver Diseases, St. Clara Hospital and University Hospital Basel, CH-4058 Basel, Switzerland

**Keywords:** colorectal cancer (CRC), muscarinic acetylcholine receptor M3 (M3R) expression, human colorectal cancer survival, tissue microarray, immune cell markers, TIA-1, therapeutic target

## Abstract

Muscarinic acetylcholine receptor M3 (M3R) has repeatedly been shown to be prominently expressed in human colorectal cancer (CRC), playing roles in proliferation and cell invasion. Its therapeutic targetability has been suggested in vitro and in animal models. We aimed to investigate the clinical role of MR3 expression in CRC for human survival. Surgical tissue samples from 754 CRC patients were analyzed for high or low immunohistochemical M3R expression on a clinically annotated tissue microarray (TMA). Immunohistochemical analysis was performed for established immune cell markers (CD8, TIA-1, FOXP3, IL 17, CD16 and OX 40). We used Kaplan–Meier curves to evaluate patients’ survival and multivariate Cox regression analysis to evaluate prognostic significance. High M3R expression was associated with increased survival in multivariate (hazard ratio (HR) = 0.52; 95% CI = 0.35–0.78; *p* = 0.001) analysis, as was TIA-1 expression (HR = 0.99; 95% CI = 0.94–0.99; *p* = 0.014). Tumors with high M3R expression were significantly more likely to be grade 2 compared to tumors with low M3R expression (85.7% vs. 67.1%, *p* = 0.002). The 5-year survival analysis showed a trend of a higher survival rate in patients with high M3R expression (46%) than patients with low M3R expression CRC (42%) (*p* = 0.073). In contrast to previous in vitro and animal model findings, this study demonstrates an increased survival for CRC patients with high M3R expression. This evidence is highly relevant for translation of basic research findings into clinically efficient treatments.

## 1. Introduction

Colorectal cancer (CRC) is one of the most prevalent cancer types, causing approximately 1.15 million new cases and 577,000 deaths globally in 2020 [1]. Its incidence in patients aged 20–50 is observably increasing, particularly in the left-sided colon and rectum [2,3]. Concurrently, a rise in high-risk and metastasized (Union for International Cancer Control (UICC) stages II, III and IV) early-onset CRC cases can be observed [4], which require especially precise and efficient adjuvant therapies [5]. However, the selection of adjuvant therapy regimens is currently mainly based on criteria such as tumor extent, tumor grade, lymph-node status, and lymphatic and venous invasion [6], which are not sufficient to describe tumor aggressiveness, prognosis and targetability [6,7]. Hence, further characterization is needed to increase prognostic predictability and to provide new targets for improved therapies [8,9]. While the number of potential prognostic markers is growing, their clinical role often remains unclear [10,11,12,13,14]. Recently, through advanced techniques such as gene and proteome analysis, many new biomarkers of CRC have been suggested in vitro, for example, immune checkpoint molecules such as OX40, receptors such as CXCR4 and CX3CR1 and kinases such as FJX1 (four-jointed box kinase 1), as well as micro-RNAs [10,13,15,16,17,18,19]. Data on the respective clinical expression and efficacy of these biomarkers is lacking. Muscarinic receptor subtype M3 (M3R) has been described as a promotor of cell proliferation in CRC and may serve as a new prognostic and predictive marker [20]. Muscarinic acetylcholine receptors are G-protein-coupled receptors comprising five subtypes (M1-M5), which correspond to the genes CHRM1-5 [20]. Of the known muscarinic receptor subtypes, M3 has been shown to be expressed exclusively in the HT29 colon cancer cell line, which suggests its potential benefit for prognosis and therapy for human CRC [21]. Furthermore, muscarinic receptor antagonists were reported to inhibit unstimulated H508 colon cancer cells by approximately 40%, while acetylcholinesterase inhibitors increased proliferation by 2 to 2.5-fold [22]. The effects of M3R activation in CRC tissue are seemingly not limited to proliferation but may also play an important role in cell invasion. Acetylcholine increased the expression of matrix metalloproteinase 1 (MMP-1) and stimulated the invasion of HT29 and H508 colon cancer cells into human umbilical vein endothelial cell monolayers [23]. All these findings suggest that the overexpression of M3R may lead to an increase in proliferation and invasiveness in CRC. We recently demonstrated the efficacy of MR3 inhibition by darifenacin in vitro and in vivo via a CRC xenograft mouse model [24]. 

However, data about the prognostic significance of M3R expression in CRC are scarce. In particular, its role in human CRC remains unclear. The goal of this study was to assess the prognostic significance of M3R expression in human CRC and its correlation with established prognostic immune cell markers on the basis of the findings of our previous publication [24].

## 2. Results

Results are presented following the reporting recommendations for tumor marker prognostic studies (REMARK) [25].

### 2.1. Clinicopathological Patient Characteristics

Tissue samples of 754 patients with CRC were analyzed. The median age was 70 years (range: 30–96) (Table 1). A total of 407 patients were female, and 347 were male. The mean tumor size was 50.8 mm, with a range of 5–170mm. Tumor location was the left hemicolon in 520 cases and the right hemicolon in 232 cases. Of all cases, 288 were rectal cancer. A total of 97 cases were UICC stage I, 288 cases were UICC stage II and 342 cases were UICC stage III. The tumor border configuration was infiltrative in 516 specimens and pushing in 221. Vascular invasion was present in 207 specimens and not present in 532. The TMA contained 658 mismatch repair (MMR)-proficient specimens and 96 MMR-deficient specimens.

### 2.2. Association of M3 Low and High Expression with Clinicopathological Features in CRC

Clinicopathological features in CRC under examination in this study and their relation to the two subgroups of low and high M3R expression samples are shown in Table 2. After immunohistochemical processing, a total of 635 punches remained for the evaluation of M3R expression (Figure 1). Of these, 568 showed a high expression of M3R, and 67 showed a low expression of M3R (Figure 2). 

CRC tissues with a high M3R expression were significantly more likely to be of the non-mucinous histologic subtype as compared to the mucinous subtype than specimens with a low M3R expression (Table 2). CRC specimens with high M3R expression were also more likely to be tumor grade G2 (85.7% of specimens with high M3R expression) compared to CRC with low M3R expression (67.1% of specimens with low M3R expression). Specimens with low M3R expression had a higher proportion of tumor grade G3 (14.9%) compared to specimens with a high M3R expression (5.1%) (Table 2).

### 2.3. Immune Cell Density According to M3 Low and High Expression

We further tested for immune cell infiltration with different well-established immune cell markers in CRC and their relation to the two subgroups of low and high M3R expression (Table 3). In patients with a low expression of M3R, a significantly higher density of CD8 and TIA-1-positive immune cells was observed (Table 3). No correlation was found for FOXP3, IL 17, CD16 and OX 40.

### 2.4. Survival Analysis

The mean overall survival time was 58.9 months (range 1–152 months). The five-year survival rate was 45% (95% CI = 49.8–57.4). The 5-year survival rate for patients with high M3R expression (46%) showed a trend of higher survival than patients with low-M3R-expression CRC (42%) (*p* = 0.073) (Figure 3).

### 2.5. Uni- and Multivariate Cox Regression Survival Analysis of Low and High Expression of M3R

In univariate Cox regression survival analysis, high expression of M3R (hazard ratio (HR), 0.73; 95% CI, 0.52–1.03; *p* = 0.075) and CD8 (HR per immune cell, 0.99; 95% CI, 0.98–1.0; *p* = 0.014) were associated with increased survival, whereas male gender (HR, 1.28; 95% CI, 1.05–1.56; *p* = 0.015), age > 60 (HR, 1.03; 95% CI, 1.02–1.04; *p* < 0.001), vascular invasion (HR, 2.57; 95%, CI 2.09–3.16; *p* < 0.001), invasive margin configuration (HR, 2.02; 95% CI, 1.58–2.59; *p* < 0.001), MMR proficiency (HR, 1.61; 95% CI, 1.15–2.56; *p* = 0.005), higher T stage (HR, 3.02; 95% CI, 19–4.35, *p* < 0.001) and lymph-node positivity (HR, 2.82; 95% CI, 2.28–3.47; *p* < 0.001) were associated with worse survival (Table 4).

In multivariate Cox regression survival analysis, high M3R expression was significantly associated with a risk for increased survival (HR, 0.52; 95% CI, 0.35–0-78; *p* = 0.001), as was TIA-1 expression (HR per immune cell = 0.97; 95% CI = 0.94–0.99; *p* = 0.014). In contrast, male sex (HR, 1.35; 95% CI, 1.05–1.74; *p* = 0.017), age > 60 (HR, 1.04; 95% CI, 1.03–1.05; *p* < 0.001), vascular invasion (HR, 2.03; 95% CI, 1.55–2.66; *p* < 0.001), MMR proficiency (HR, 1.59; 95% CI, 1.01–2.49; *p* = 0.043), higher T stage (HR, 2.30; 95% CI, 1.47–3.60; *p* < 0.001) and higher N stage (HR, 2.26; 95% CI, 1.72–2.96; *p* < 0.001) were associated with a risk for poorer survival (Table 4).

Neither univariate nor multivariate Cox regression showed any significance for the independent impact of M3R on overall survival.

## 3. Discussion

### 3.1. Key Findings

We aimed to examine the potential of M3R expression as a prognostic factor for survivability in human CRC patients, as its expression and inhibition have shown therapeutic potential in animal and in vitro trials [24]. We found that high M3R expression correlated with increased survivability, significantly with lower tumor grade and non-mucinous subtype, associating with a more favorable outcome as compared to low M3R expression, which was significantly correlated with decreased survivability, higher tumor grade and mucinous subtype.

### 3.2. Correlation with Previous Literature

These findings are surprising and stand in contrast to the previously reported in vitro and in vivo effects of M3R expression and inhibition [20,26,27,28,29,30]. Other examples of a strong inverse correlation between tumor grade and receptor expression are described in the literature, for example, numerous studies concerning breast cancer tumor grade and hormone receptor intensity [31,32,33,34], and in astrocytomas, estrogen receptor expression was positive only in low-grade and nil in high-grade astrocytomas [35]. A progressive decrease in progesterone receptor and estrogen receptor 1 mRNA expression was observed from endometrioid endometrial cancers to more aggressive serous tumors as defined by grade level [36]. Pacini et al. examined the expression of muscarinic acetylcholine receptor subtypes M1, M2 and M3 in transitional cell carcinoma of the bladder and found that M1R and M3R were significantly upregulated only in low-grade samples [37]. The correlation between tumor grade and M3R expression in CRCs warrants further investigation. 

Aiming to discover further potential treatment approaches, we investigated the role of MR3 expression in the CRC tumor immune microenvironment by immunohistochemical analysis for a selection of promising immunomarkers: CD8, FOXP3, IL17, TIA-1, CD16 and OX40. Each of these have previously been shown to play a role in CRC progression by fellow researchers [16,38,39,40,41,42,43] and members of our group [11,13,14,44]. Immune cell density of the examined CRC specimens showed that low M3R expression was significantly associated with a higher density of CD8- and TIA-1-positive immune cells. A possible link between increased density of CD8-positive cells in CRC tumors and their level of differentiation has been shown [45]. Sun-Young Lee et al. found that CD8+ T cell infiltration in the tumor stroma was more prominent in moderately and poorly differentiated adenocarcinoma than in adenoma and well-differentiated adenocarcinoma [46]. Similar associations were found in breast cancer, where tumor-infiltrating CD8-positive T cells significantly increased with stage progression [45]. In our study, a higher density of CD8-positive cells was significantly associated with low M3R expression. There is also increasing evidence that a higher density of CD8- and TIA-1-positive immune cells in CRC tumors is a prognostic factor for increased survival [46]. These findings highlight the incomplete picture of interactions between tumor grade, immune cell density, M3R expression and survival.

The overexpression of M3R has been shown in the HT-29 human colorectal adenocarcinoma cell line through subtype-specific muscarinic antagonists and X-ray microanalysis measurement of intracellular ion concentration [21,47]. Prior research has shown that surgical CRC samples may exhibit increased M3R expression by up to 128-fold in 10 out of 18 specimens as compared to an adjacent normal colon [26]. The possible effects of M3R activation in CRC have been studied in vitro using H508 human colon cancer cells, suggesting that muscarinic receptor agonists stimulate cell proliferation, migration and invasion by several post-M3R signaling pathways, one example being acetylcholine-stimulated calcium-dependent phosphorylation of p44/42 mitogen-activated protein kinase (MAPK) [26,27]. Raufman et al. used an animal model employing Apc^min/+^ mice to compare Chrm3^+/+^ mice (capable of M3R expression) to Chrm3^-/-^ mice (not capable of M3R expression) and showed a 70% reduction in the number of tumors and an 81% reduction in tumor volume in the group that was not capable of M3R expression [28].

These in vitro and animal trials are part of an emerging body of evidence pointing to M3R expression rate as a biomarker for increased proliferation and invasiveness of CRC, as recently shown by Hering et al. [24].

Apart from CRC, M3R has clearly been shown to play a role in lung cancer [48,49,50]. Additionally, muscarinic agonists have been reported to have the ability to stimulate growth for melanoma, pancreatic, breast, ovarian, prostate and brain cancers [50,51,52]. Thus, these cancer types need to be considered in the context of our study and results in the future.

### 3.3. Implications

Our findings after examining MR3 expression in surgical samples of more than 600 human CRC patients suggest that while in in vitro and animal model studies, there is considerable potential for M3R inhibition, this may not be the case for clinical treatments. This may either be due to variables that are not present in a laboratory setting or to differences in laboratory variables such as the type of the investigated molecule (e.g., micro-RNA or protein), the type of antibody used, or differences in staining techniques and scoring systems.

CRC cells with high M3R expression could have traits that increase overall survival that are unrelated to tumor proliferation and invasion. An animal trial using Chrm3^-/-^ mice showed that genetic ablation of M3R affected mucus production by decreasing mucin 2 gene expression, thereby facilitating prolonged bacterial adherence and delaying clearance of C. rodentium [53]. CRCs with a low expression rate of M3R could therefore have higher tendencies for bacterial infection. Muscarinic receptor activation on colon epithelial cells has been shown to protect against cytokine-induced barrier dysfunction by inhibiting IL-1β-induced production of chemokines and rearrangement of tight-junction proteins, while this protective effect of acetylcholine was antagonized by atropine [54]. This effect may lead to a stronger inflammatory response and higher inflammation rates in CRCs with low M3R expression.

Cheng et al. used immunohistochemistry to identify the expression of choline acetyltransferase (ChAT), a critical enzyme for acetylcholine synthesis, in surgical specimens of normal colons and colon cancer and found that normal colon enterocytes showed limited to no ChAT staining, whereas one-half of the colon cancer specimens displayed moderate to strong staining, and the other half exhibited weak staining [22]. Despite the small sample size, this suggests a higher rate of ACh production in colon cancer cells. The correlation between high and low M3R expression and ACh production capability of CRC cells in surgical specimens should be investigated. 

Experiments have shown a relationship between the M3R expression rate and M3R inhibition or activation in different cell types. Witt-Enderby et al. found that in rabbit bronchi, M2R and M3R were significantly upregulated compared to the control after a 4-week inhibition by atropine [55]. A similar observation was made in rat forebrains by Wall et al., where a 14-day administration of atropine resulted in a 69% increase in the density of M3R [56]. Fukamauchi et al. studied the administration of carbachol, a cholinergic agonist, to cerebellar granule cells and described a time-dependent loss of M3R mRNAs as a result of stimulation [57]. Further experiments showed a decrease of 59.3% in M3R gene transcription in nuclei from cells treated with carbachol and a 230% increase in M3R gene transcription in nuclei from cells treated with atropine [57].

While the retrospective nature of this study is a limitation, our data contribute to the development of targeted, prospective studies in the future. Additionally, the investigated cohort includes CRC patients who underwent surgery between 1985 and 1998, a period in which neoadjuvant therapy regimens had not yet been widely established. Thus, while our results may not represent the efficacy of current clinical treatments fully, they are more likely to portray CRC immunobiology accurately due to the absence of the effect of antineoplastic agents.

We found that CRC tissues with a high M3R expression were significantly more likely to be of the non-mucinous histologic subtype as compared to the mucinous subtype than specimens with a low M3R expression (Table 2). Patients with a mucinous adenocarcinoma (MAC), which we found to be correlated with low M3R expression, were reported to be younger by Kanemitsu et al., have greater disease severity and metastatic spread and a significantly shorter 5-year survival rate than patients with a non-mucinous subtype CRC. The association of low M3R expression with the MAC subtype could be one reason for the decreased survival in the low-M3R group, which requires further investigation.

### 3.4. Future Perspective and Possibilites

Our findings add an important perspective for future trials to an abundant and currently rapidly increasing number of in vitro genetic and proteomic findings in CRC. In an era of high-throughput screening, next-generation sequencing and proteome-specific therapeutic agents leading to personalized cancer therapies, the inclusion of the tumor (immune) microenvironment is paramount when translating molecular findings into suitable therapies and biomarkers. Thus, we believe that our findings should be investigated in advanced cancer models. Delle Cave et al. already reviewed promising 3D in vitro cancer models for pancreatic cancer [58]. Using such a model to screen for M3R expression and associated tumor cell and microenvironment interactions mechanistically would ultimately generate new treatment options. The first preclinical 3D models for CRC have been proposed [59], but many more are needed. Additionally, M3R signaling could be investigated by using nanoparticles in vitro and in vivo, which have been shown to deliver small molecules specifically to CRC cells [60]. The structure–activity relationships of muscarinic receptor subtypes and the therapeutic effects of novel M3R-specific ligands or modulators also need to be explored [29,30,61]. Tolaymat et al. reported that M3R deletion increased proliferation in intestinal stem cells and that M3R expression fine-tuned the cellular response to acetylcholine stimulation, ensuring intestinal tissue homeostasis [20]. In connection with our findings, this warrants further investigation. More established immune markers need to be investigated, such as interferon-gamma, CTLA-4 and CD28, to further characterize the tumor immune microenvironment. Further mechanisms to be explored in order to examine the role of M3R signaling in CRC could include micro-RNAs, members of the CHRM3-dependent oncogenetic pathways and potentially synthetically lethal combinations with M3R signaling members [17,18,62,63,64]. Future trials may incorporate proteomic screening [65], multiplex immunofluorescence [66] and screening of The Cancer Genome Atlas (TCGA; https://www.cancer.gov/tcga, accessed on 24 April 2023) for the impact of M3R expression on survival in both healthy and CRC patients. Appropriate gene and protein microarrays, bioinformatics and artificial-intelligence-based screening models could also be implemented in future investigations of M3R expression [15,19,67,68].

## 4. Materials and Methods

### 4.1. Tissue Microarray (TMA) Construction

A tissue microarray (TMA) was constructed at the Department of Pathology, University Hospital Basel, from each tissue sample from 754 unselected, non-consecutive patients with primary CRC following approval by the Regional Ethical Committee (EKBB, Ethikkommission beider Basel, Switzerland). Formalin-fixed, paraffin-embedded tissue blocks were prepared according to standard procedures. Tissue cylinders with a diameter of 0.6 mm were punched from morphologically representative areas of each donor block and brought into one recipient paraffin block (30 × 25 mm) using a TMA-Grand Master^®^ automated tissue arrayer (3DHisteck, Sysmex AG, Horgen, Switzerland). Each punch was made from the center of the tumor so that each TMA spot consisted of at least 50% tumor cells. The detailed construction technique was previously described by our group [44,69,70].

### 4.2. Clinicopathological Features

Clinicopathological data for the 754 included CRC patients were collected retrospectively in a non-stratified and non-matched manner. Annotation included patient age, pT/pN stage, grade, histologic subtype, tumor location, diameter, vascular invasion, border configuration, presence of peritumoral lymphocytic inflammation at the invasive tumor front and overall survival. After microscopy and storage using a ZEISS Axio Scan.Z1 slide scanner, tumor border configuration and peritumoral lymphocytic inflammation were evaluated using the original H&E slides of the resection specimens corresponding to each tissue microarray punch [67]. Available follow-up data for the testing and validation cohort had a mean event-free follow-up time of 115 and 36 months, respectively. 

### 4.3. Immunohistochemistry

Immunohistochemical staining was performed using an anti-M3R primary antibody (1:100, AHP1355, Biorad Laboratories, Neuberg, Germany) on a Benchmark immunohistochemistry staining system (Leica Biosystems, Muttenz, Switzerland) with bond polymer refine detection solution (Leica Biosystems) for 3, 3′-diaminobenzidine. Antigen retrieval was performed using citrate solution at pH 6 for 30 min at 95 °C. M3R staining intensity was scored from 0 (no reaction) to 3 (strong reaction) for each TME punch. Low expression of M3R was defined as scores of 0 and 1, and high expression was defined as scores of 2 and 3. Scoring was performed by two trained research fellows (M.S. and M.A.), and data were independently validated by an additional investigator (B.W.). Expression of M3R was scored according to the staining intensity.

### 4.4. Statistical Analysis

M3R-positive cells were counted on each of the 635 CRC TMA cores. After having proven an association between M3R-positive cells and overall survival (OS) by univariate Cox regression, an optimal threshold was estimated by regression tree analysis. The obtained threshold was found to be almost equal to the 25th-percentile value. Therefore, continuous values were dichotomized, subdividing the collective as CRC with low or high M3R expression. Chi-Square or Fisher’s exact tests were used to determine the association of M3R expression and clinicopathological discrete features, and Wilcoxon’s signed-rank sum test was used for comparison with continuous values. Survival curves were depicted according to the Kaplan–Meier method and compared with the log-rank test results. Only tests for normal and non-normal distribution were used (Mann–Whitney U). Age and tumor size were evaluated using the Kruskal–Wallis test. Normally distributed data were presented with mean ± SD, and non-normally distributed data were presented with median (range).

The immune cells CD8, FOXP3, IL17, TIA-1, CD16 and OX40 have previously been evaluated for CRC, and CRC cases have been classified accordingly by our group and others [1,2,3,4,5,6,7,8,9,10,11], generating the data used in the present study. The assumption of proportional hazards was verified for all markers by analyzing the correlation of Schoenfeld residuals and the ranks of individual failure times. Any missing clinicopathological information was assumed to be missing at random. 

M3R expression data were entered into multivariate Cox regression analysis, and hazard ratios (HRs) and 95% confidence intervals (CIs) were used to determine prognostic effects on survival time. For the immune cell biomarkers, the hazard ratio was calculated per immune cell, subsequently indicating that, e.g., 10 additional immune cells in a sample would generate an HR of 0.991010 = 0.9 or, in the case of 20 more cells, 0.992020 = 0.82. 

Additionally, the independent impact of M3R on overall survival was investigated by univariate and multivariate Cox regression. 

All *p*-values were two-sided and considered significant when *p* < 0.05. Analyses were performed using STATA 13 (StataCorp, College Station, TX, USA).

## 5. Conclusions

While our knowledge of CRC is increasing, new questions are also constantly arising, which must be investigated scientifically to improve treatments [37]. Our data offer several new insights that may help to navigate future investigations. This study is highly relevant, as it points out an important, unexpected difference in the prognostic and therapeutic role of M3R expression in CRC between laboratory and clinical settings. Evidently, there are in vivo factors influencing the expression and activation of M3R and its effect on survivability that need to be explored to reveal the broader picture and further utilize M3R expression for therapeutic advances in the treatment of CRC.

## Figures and Tables

**Figure 1 ijms-24-08198-f001:**
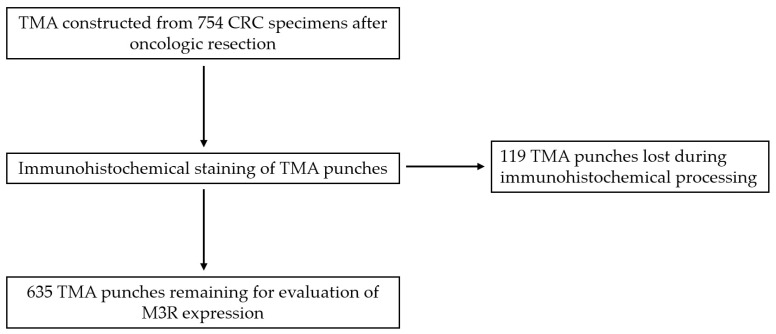
Transition from total number of tissue samples to remaining number of tissue microarrays (TMAs) after immunohistochemical processing. Abbreviations: TMA = tissue microarray; CRC = colorectal cancer.

**Figure 2 ijms-24-08198-f002:**
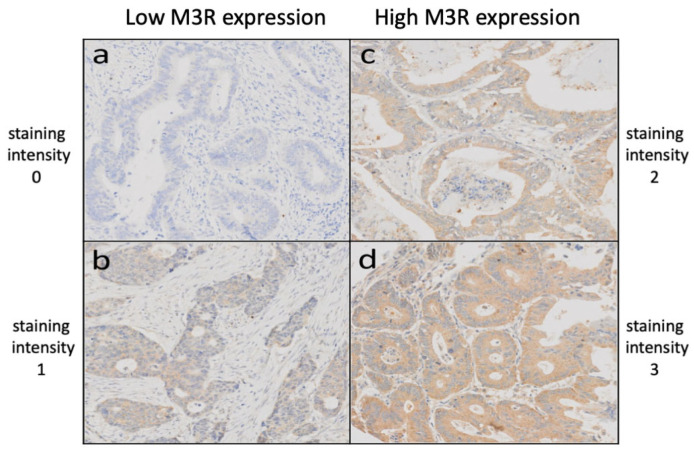
Staining intensities of CRC TMAs after immunohistochemical staining using an anti-M3R primary antibody, revealing low M3R expression (**a**,**b**) and high M3R expression (**c**,**d**). Abbreviations: TMA = tissue microarray; CRC = colorectal cancer; M3R = muscarinic acetylcholine receptor M3.

**Figure 3 ijms-24-08198-f003:**
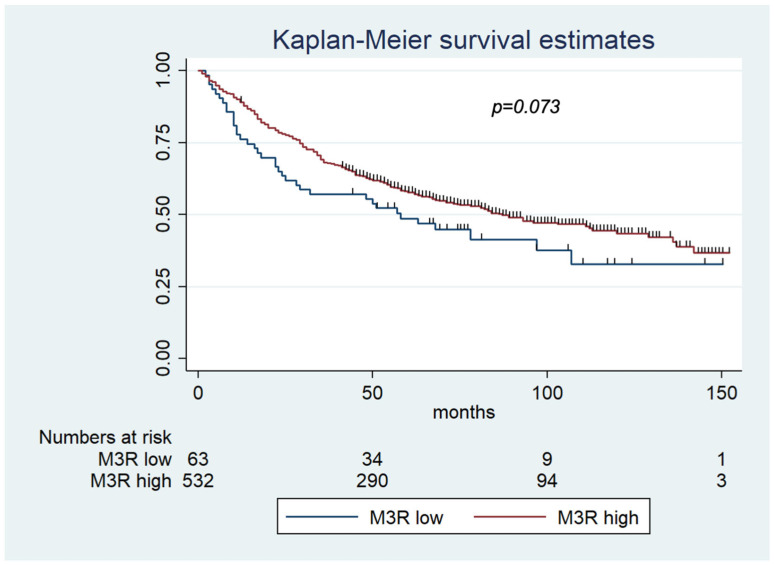
Kaplan–Meier survival estimates for low MR3 expression and high MR3 expression. The 5-year survival rate showed a trend of higher survival for patients with high-M3R CRC (46%) than patients with low-M3R CRC (42%) (*p* = 0.073). Abbreviations: M3R-low = CRC specimens with low M3R (muscarinic acetylcholine receptor M3) expression; M3R-high = CRC specimens with high M3R (muscarinic acetylcholine receptor M3) expression.

**Table 1 ijms-24-08198-t001:** Clinicopathological characteristics of the CRC patient cohort ^1,2^.

Patient Characteristic		N or Mean	Percentage or Range
Age, years (median, mean)		70, 68.8	30–96
Tumor size in mm (median, mean)		50, 50.8	5–170
Sex	Female	407	50%
	Male	347	43%
Anatomic site of the tumor	Left-sided	520	64%
	Right-sided	232	29%
	Stage IA, T1N0	26	3%
UICC stage	Stage IB, T2N0	71	9%
	Stage IIA, T3N0	254	31%
	Stage IIB-C, T4N0	34	4%
	Stage III, >N0	342	42%
Tumor border configuration	Infiltrative	516	63%
	Pushing	221	27%
Vascular invasion	No	532	65%
	Yes	207	25%
Microsatellite stability	Proficient	658	81%
	Deficient	96	96%
Rectal cancers		288	35%
Rectosigmoid cancers		50	6%
Overall survival time (months)		58.9	1–152
5-year survival % (95% CI)		0.45	0.42–0.49

^1^ Percentages may not add up to 100% due to missing values for some variables. ^2^ Abbreviations: N = total number of observations; UICC = Union for International Cancer Control; T = size or extension of primary tumor; N = degree of spread to regional lymph nodes; M = presence of distant metastasis.

**Table 2 ijms-24-08198-t002:** ^1,2^: Association of M3 low and high expression with clinicopathological features in CRC.

Parameter			M3R-Low		M3R-High	*p*-Value
			N = 67 (10.6%)		N = 568 (89.5%)	
Age	Years, mean ± SD		67.3 ± 11.9		68.8 ± 11.4	0.418
Tumor diameter	mm, mean ± SD		50.5 ± 22.1		50.9 ± 20.1	0.760
Gender	Female		33 (49.3%)		294 (51.7%)	0.640
	Male		31 (46.3%)		244 (42.9%)	
Tumor location	Left-sided		42 (62.7%)		379 (66.7%)	0.401
	Right-sided		22 (32.8%)		157 (27.6%)	
Histologic subtype	Mucinous		6 (9.0%)		22 (3.9%)	**<0.001**
	Non-mucinous		61 (91.0%)		546 (96.1%)	
pT stage	pT1–2		14 (20.9%)		105 (18.5%)	0.436
	pT3–4		44 (65.7%)		425 (74.8%)	
pN stage	pN0		32 (47.8%)		279 (49.1%)	0.690
	pN1–2		31 (46.3%)		243 (42.8%)	
Tumor grade	G1		3 (4.5 %)		14 (2.5%)	**0.002**
	G2		45 (67.1%)		487 (85.7%)	
	G3		10 (14.9%)		29 (5.1%)	
Vascular invasion	Absent		39 (58.2%)		387 (68.1%)	0.350
	Present		19 (28.4%)		143 (25.2%)	
Tumor border	Pushing		12 (17.9%)		171 (30.1%)	0.068
	Infiltrating		46 (68.7%)		357 (62.9%)	
PTL inflammation	Absent		47 (70.2%)		406 (71.5%)	0.446
	Present		11 (16.4%)		124 (21.8%)	
Microsatellite stability	Deficient		8 (11.9%)		60 (10.6%)	0.747
	Proficient		56 (83.6%)		478 (84.2%)	
5-year survival rate	(95% CI)		42% (0.30–0.54)		46% (0.41–0.50)	0.073

^1^ Percentages may not add up to 100% due to missing values of some variables; age and tumor size were evaluated using the Kruskal–Wallis test. Gender, anatomical site, T stage, N stage, grade, vascular invasion and tumor border configuration were analyzed using the χ² test. Survival analysis was performed using the Kaplan–Meier method. ^2^ Abbreviations: M3R-low = CRC specimens with low M3R (muscarinic acetylcholine receptor M3) expression; M3R-high = CRC specimens with high M3R (muscarinic acetylcholine receptor M3) expression; N = total number of observations; SD = standard deviation; mm = millimeters; pT = histopathological size or extension of primary tumor; pN = histopathological degree of spread to regional lymph nodes; G = tumor grade; PTL inflammation = peritumoral lymphocytic inflammation.

**Table 3 ijms-24-08198-t003:** ^1^ Immune cell density according to M3 low and high expression.

	M3-Low	M3-High	*p*-Value
	Mean ± SD	Mean ± SD	
CD8	18.9 ± 32.6	8.67 ± 16.8	**0.018**
FOXP3	30.3 ± 34.9	34.0 ± 39.4	0.638
IL17	10.3 ± 15.6	15.8 ± 29.1	0.141
TIA-1	4.5 ± 8.0	2.7 ± 6.9	**0.047**
CD16	30.0 ± 32.1	32.9 ± 32.1	0.292
OX40	37.3 ± 49.0	45.5 ± 60.5	0.380

^1^ Abbreviations: CD8= cluster of differentiation 8; FOXP3 = forkhead box P3; IL17 = interleukin 17 family; TIA-1 = TIA1 cytotoxic granule-associated RNA-binding protein; CD16 = cluster of differentiation 16; OX40 = tumor necrosis factor receptor superfamily, member 4.

**Table 4 ijms-24-08198-t004:** ^1^ Uni- and multivariate Cox regression analysis.

Variable	Univariate Regression		Multivariate Regression	
	OR (95% CI)	*p*-Value	OR (95% CI)	*p*-Value
Sex				
Female	Reference		Reference	
Male	1.28 (1.05–1.56)	0.015	1.35 (1.05–1.74)	0.017
Age				
<60	Reference		Reference	
>60	1.03 (1.02–1.04)	<0.001	1.04 (1.03–1.05)	<0.001
Vascular invasion				
Absent	Reference		Reference	
Present	2.57; 2.09–3.16;	<0.001	2.03 (1.55–2.66)	<0.001
Invasive tumor margin configuration				
Pushing	Reference		Reference	
Infiltrative	2.02 (1.58–2.59)	<0.001	1.34 (0.96–1.84)	0.07
Microsatellite stability				
Deficient	Reference		Reference	
Proficient	1.61 (1.15–2.56)	0.005	1.59 (1.01–2.49)	0.043
pT Stage				
pT 1–2	Reference		Reference	
pT 3–4	3.09 (2.19–4.35)	<0.001	2.30 (1.47–3.60)	<0.001
pN Stage				
pN 0	Reference		Reference	
pN > 0	2.82 (2.28–3.47)	<0.001	2.26 (1.72–2.96)	<0.001
Grade				
G0-G1	Reference		Reference	
G2-G3	6.06. (1.94–18.87)	0.002	2.58 (0.62–10.80)	0.193
M3R expression				
low	Reference		Reference	
high	0.73 (0.52–1.03)	0.075	0.52 (0.35–0.78)	0.001
CD8 expression				
low	Reference		Reference	
high	0.99 (0.98–1.0) ^2^	0.014	1.00 (0.99–1.01) ^2^	0.609
TIA-1 expression				
Low	Reference		Reference	
High	0.99 (0.98–1.0) ^2^	0.24	0.97 (0.94–0.99) ^2^	0.014

^1^ Abbreviations: OR = odds ratio; CI = confidence interval; pT = histopathological size or extension of primary tumor; pN = histopathological degree of spread to regional lymph nodes; G = tumor grade; M3R = M3R (muscarinic acetylcholine receptor M3) expression of CRC specimens; CD8 = cluster of differentiation 8; TIA-1 = TIA1 cytotoxic granule-associated RNA-binding protein. ^2^ Odds ratio calculated per immune cell, subsequently indicating that, e.g., 10 additional immune cells in a sample would generate an HR of 0.991010 = 0.9 or, in the case of 20 more cells, 0.992020 = 0.82.

## Data Availability

The data presented in this study are available on request from the corresponding author. The data are not publicly available due to patient data protection and the decision of the regional ethical committee at the time of study approval.

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
