# Peer review of "Muscarinic Acetylcholine Receptor M3 Expression and Survival in Human Colorectal Carcinoma—An Unexpected Correlation to Guide Future Treatment?"

_ijms, 2023, doi:10.3390/ijms24098198_

Round 1

Reviewer 1 Report

The manuscript by Lobbes and co-workers addresses an interesting and pertinent topic, since it is relevant to establish human colorectal cancer prognostic markers, thereby optimizing prognosis and treatment approaches. This is further underlined by the high incidence of this type of cancer.

This manuscript reports a well-reasoned clinical study, analyzing a remarkable number of patients/tissue samples and a comprehensive panel of immunocell markers. The results have potential applicability in the clinical settings, which represents an added value. The draft is well structured.

Despite its current flaws, which I detail below, I believe that this paper fits the scope of the IJMS.

General remarks

·      I recommend that the manuscript is revised for formatting, since some aspects, namely text alignment and formatting, as well as the way how some numbered citations were introduced, do not fully comply with those of the IJMS template.

Keywords

·      I suggest the keywords to be simplified/shortened (e.g., “tissue microarray for colorectal cancer” -> “tissue microarray” only, since “colorectal cancer” already is another keyword).

Results

·      Section 2.1: Please substantiate the difference between the number of CRC patients said to have been analyzed in the Abstract (n = 814) and in the Discussion (n = 754) sections.

Discussion

·      Please better substantiate the selection of immune cell markers analyzed in the study.

·      Please acknowledge future prospects/opportunities for complementary assays. For instance, additional biomarker genes and proteins (and the respective analyzing techniques) can be suggested.

Author Response

                                                                            Berlin, April 17th, 2023

Dear Reviewer,

in the attached Word file you will find the revisions of the manuscript Muscarinic Acetylcholine Receptor 3 expression and survival in human colorectal carcinoma – an unexpected correlation to guide future treatment? to be considered for publication as an original article.

We found the reviewers` comments very helpful and believe that these comments and our revisions have further improved the quality of this manuscript.

The requested changes are listed in the response documents and will additionally be visible as “tracked-changes” in the re-submitted manuscript.

All authors of this research paper have directly participated in the planning, execution, and/or analysis of this study. The revised manuscript has been seen and approved by all authors.

Thank you for your time and attention to our revisions. We hope you will find our manuscript suitable for publication and look forward to hearing from you.

Yours sincerely, on behalf of all authors

Dr. med. Leonard A. Lobbes
Charité - Universitätsmedizin Berlin,
Department of General and Visceral Surgery
Hindenburgdamm 30, 12203 Berlin, Germany
Phone: 0049 030 450 622 789 / E-Mail: [email protected]

Reviewer 2 Report

Introduction. There is a significant gap in the flow of the text in the first paragraph and the passages therein are not connected fully, if at all. Please modify the section accordingly to make it easier for readers.

Statistical analysis. I have two comments. First: the authors have not made full use of the findings of the study, see comment in the results, second: the statistics are described badly, for example, the Cox technique and the modeling for the multivariable analysis are cited in the results but not in the methodology. This is an important omission.

Results.

The manuscript lacks visualization of the results. The authors only included tables and omitted figures, which would help readers get a better grasp of the findings.

Also, the authors have not evaluated the findings, and have not made as many associations as possible.

These two points can be improved.

Discussion.

I suggest to divide the discussion in two or three smaller sub-sections.

Author Response

(The authors gave the same response as above.)

Reviewer 3 Report

This manuscript entitled "Muscarinic Acetylcholine Receptor 3 expression and survival in human colorectal carcinoma an unexpected correlation to guide future treatment?" by Lobbes LA. et al., indicated relationship between M3R expression and survival for CRC patients. This study is very important for understanding of this pathological mechanisms. But some corrections may be needed. In introduction section,  it was better to add some reports previously published as for bomarkers and molecular pathways of CRC. In Table, it was better to add a n expression  dara of interferon-gamma, CTLA-4, and CD28. In 5. conclusion section, it was better to clear the novel findings in this study.

Author Response

(The authors gave the same response as above.)

Reviewer 4 Report

In the submitted manuscript authors assessed prognostic significance of muscarinic acetylcholine receptor M3 (M3R) expression in colorectal carcinoma (CRC) and found that higher M3R expression is an independent prognostic biomarker for better (overall or disease-specific ?!) survival of CRC patients.

Although this is a single biomarker study, author used quite a large number of samples, so observed results could be credible. However, this manuscript has many drawbacks, especially related to presentation of the results and dubious application of statistical analyses, so in present form it does not meet the standards of a typical biomarker discovery/validation report (please inspect REMARK guidelines https://www.equator-network.org/reporting-guidelines/reporting-recommendations-for-tumour-marker-prognostic-studies-remark/).

Therefore, authors should correct and further improve their manuscript:

1) Inspect papers like https://doi.org/10.3390/life12101559 and accordingly organize your manuscript in and way that order of the results should be:

- presentation of basic clinicopathological characteristics (CPCs) of patients from whom tumor samples were taken

- association of M3R with CPCs

- correlation between M3R and "established immunocell markers (CD8, TIA-1, FOXP3, IL 17, CD16 and OX 40)"

- impact of ALL studied biomarkers and CPCs on survival (Kaplan-Meier curves and log-rang test)

- "independent" impact of M3R on survival (univ. and multiv. Cox prop. regression) including ALL CPCs and established immunocell markes

2) Do NOT present statistically insignificant results as significant! E.g., correct statement would be "The 5-year-survival rate was statistically insignificantly higher for patients with high M3R expression (46%) than patients with low M3R expression CRC (42%) (p=0.073).".

3) Take care when interpret (and stress that out) statistically significant HR close to 1.00, like for TIA-1 expression (HR=0.99; 95%CI=0.94-0.99; p=0.014), since this is really marginally decreased relative risk!

4) Methods should contain much more details so this study could be completely reproducible. E.g, explain how ALL markers were determined, including "established immunocell markers (CD8, TIA-1, FOXP3, IL 17, CD16 and OX 40)", state the exact models of all used instruments (like semiautomated tissue arrayer and microscope used for observing stained TMAs).

5) Re-write "4.3 Statistical analysis" so that it includes ALL used statistical tests (KM, Cox, etc.).

6) Application of statistical tests and presentation of the results is specifically dubious:

- According to 4.3 section, it seems that ad hoc parametric tests were used without first testing for normality of distribution, although in tables' footnotes Kruskal-Wallis test was mentioned?!

- Data were presented with median (range) and/or mean±SD, while in 4.3 section it is written "Descriptive statistics include mean ± standard error of the means". Normally distributed data should be presented with mean±SD or mean±SEM, and non-normally with median (range), and this MUST be uniform throughout the manuscript!

- There is no need to mention statistical tests for Table 1 since it does not present any results of statistical analyses!

- At several points authors presented overall survival time while in line 227 it is mentioned "disease-specific survival", so now it is unclear which actual endpoint was used for survival analyses!

7) Provide representative photographs of all M3R staining intensities.

8) There is no need to use numbers in superscript in tables' captions and word "Abbreviations:" in tables' footnotes.

9) Re-check manuscript that ALL abbreviations were explained after first mentioning in Abstract and main text, and in tables' and figures' captions.

10) There is no need to present number (%) and median (range) or mean±SD in two separate columns in tables, one is enough!

11) It is unclear why more complex M3R scoring system that takes into account also the number of stained cells was not used.

12) Having missing values is perfectly normal, and their numbers could be presented in tables which show basic CPCs. However, when you show data which was used for the analysis of contingency tables, i.e., chi-squared test, either vertical or horizontal total percentages sum MUST be 100%! Also, since only 635 punches were stained, it would be more meaningful to present CPCs of only those 635 patients, not all 704 (Table 1)!

13) In Table 2, it seems that 5-year survival rates should be multiplied by a hundred (e.g., 42%).

14) X-axis of KM curves should be divided in more meaningful time frames, e.g., divisible by 12 months, and figure legends should mentioned number of analyzed patients/samples.

15) Authors should throughout the manuscript use only the proper name of M3R protein: https://www.uniprot.org/uniprotkb/P20309/entry#names_and_taxonomy

16) In 'Discussion':

- More information should be provided on studies assessing prognostic significance of M3R in other types of cancers.

- Variables which ARE present in a laboratory setting should be also discusses, since studied macromolecules (mRNA or protein), used antibodies, M3A scoring system, etc. could also have an impact on discrepancies between results obtained in different studies.

16) Authors could use TCGA-CRC dataset to inspect M3R expression in CRC and healthy patients, and its impact on survival. There are many user friendly web-based tools for that, like GEPIA2, UALCAN, cBioPortal, Xena browser, etc.

Round 2

Reviewer 2 Report

Will the authors, please, add a short sub-section in the Discussion to present the clinical significance of their results?

After that, the manuscript can be accepted.

Reviewer 3 Report

This manuscript is corrected according to the reviewer's comments. I recomennd as accept.

Reviewer 4 Report

Authors have satisfactorily addressed all my concerns and substantially improved quality of their manuscript.